# Continuous Quantitative Risk Management in Smart Grids Using Attack Defense Trees

**DOI:** 10.3390/s20164404

**Published:** 2020-08-07

**Authors:** Erkuden Rios, Angel Rego, Eider Iturbe, Marivi Higuero, Xabier Larrucea

**Affiliations:** 1Tecnalia, Basque Research and Technology Alliance (BRTA), 48170 Derio, Spain; angel.rego@tecnalia.com (A.R.); eider.iturbe@tecnalia.com (E.I.); xabier.larrucea@tecnalia.com (X.L.); 2Ingeniería de Comunicaciones, Universidad del País Vasco/Euskal Herriko Unibertsitatea, 48004 Bilbao, Spain; marivi.higuero@ehu.eus

**Keywords:** information security, risk assessment, security management

## Abstract

Although the risk assessment discipline has been studied from long ago as a means to support security investment decision-making, no holistic approach exists to continuously and quantitatively analyze cyber risks in scenarios where attacks and defenses may target different parts of Internet of Things (IoT)-based smart grid systems. In this paper, we propose a comprehensive methodology that enables informed decisions on security protection for smart grid systems by the continuous assessment of cyber risks. The solution is based on the use of attack defense trees modelled on the system and computation of the proposed risk attributes that enables an assessment of the system risks by propagating the risk attributes in the tree nodes. The method allows system risk sensitivity analyses to be performed with respect to different attack and defense scenarios, and optimizes security strategies with respect to risk minimization. The methodology proposes the use of standard security and privacy defense taxonomies from internationally recognized security control families, such as the NIST SP 800-53, which facilitates security certifications. Finally, the paper describes the validation of the methodology carried out in a real smart building energy efficiency application that combines multiple components deployed in cloud and IoT resources. The scenario demonstrates the feasibility of the method to not only perform initial quantitative estimations of system risks but also to continuously keep the risk assessment up to date according to the system conditions during operation.

## 1. Introduction

Protecting Information Technologies (IT) and Operational Technologies (OT) systems from all potential risk situations and assessing their security level are two of the major challenges in industrial cybersecurity research. There is a growing need for quantifiable and demonstrable risk assessment solutions that aid in this endeavor [1]. This is particularly challenging in complex distributed industrial Internet of Things (IoT) scenarios, such as modern smart grid systems, where the attack surface spans to multiple intelligent complementary services based on IoT and private cloud resources.

These systems require holistic security approaches that support experts in managing risks in all system components, including third-party services, which often lack details of the security and privacy measures they adopt. These components are usually beyond the control of the developers and securing them is the responsibility of cloud service providers (CSPs) or IoT providers. Still, as part of the system, threats and protection in these components need to be analyzed and considered when assessing overall system risks.

In this paper, we present a solution that responds to these challenges by proposing a continuous quantitative risk management methodology for complex smart grid systems that orchestrate multiple components. The methodology is based on capturing in the form of attack-defense trees (ADTs), the envisaged attack-defense scenarios of the smart grid system. The ADTs enable reasoning on the relationships between potential attack events against different parts or components of the system. Therefore, the methodology leverages ADTs and shows how to integrate them into a single system ADT that enables an evaluation of the system risks when different attack situations are faced, and different defense strategies are adopted.

The method proposes a novel risk assessment algorithm for ADTs to propagate the risk attributes from the attack event nodes and defense nodes up to the root of the tree based on smart adversary and smart defender strategies, which would have opposite goals with respect to system risk: While the smart adversary would seek to cause the maximum risk to the system, the smart defender would try to minimize the risk.

The methodology enables the continuous reassessment of system risks after the defenses available in all the components are known, as well as continuous updates of the assessed risks when all system components are deployed and running. Hence, the methodology addresses the needs of IoT and cloud-based distributed systems, such as modern smart grids. 

Last but not least, the methodology serves for both proponent and opponent views of ADTs, that is, it enables reasoning on both attacker goals and defender goals, including scenarios where potential attacks over the defenses themselves are also possible.

The paper is structured as follows: After Section 1 introduces the paper, Section 2 offers a discussion on related works. Then, Section 3 provides all the details of the ADT-based continuous risk management methodology proposed; Section 4 describes the results of the validation of the methodology in a real smart building scenario, demonstrating the feasibility of continuous risks assessment following the approach proposed. Section 5 offers the discussion of the results together with future research lines. Finally, in Section 6, the conclusions of the work are presented.

## 2. Related Works

In the last decades, multiple graphical methods for the analysis of attack and defense scenarios have emerged. A comprehensive survey by Kordy et al. [2] is available, which compares all these formalisms. Threat logic trees introduced by Weiss in 1991 [3] pioneered the graphical attack modelling techniques. Since then, most of the literature has focused on directed acyclic graph (DAG)-based approaches mainly because they do not suffer from the state space explosion problem, which is a drawback of methods using graphs with cycles.

Two main trends can be distinguished in the field of threat analysis using directed acyclic graphs (DAGs): Models that derive from or extend attack trees (ATs), such as the one followed in this paper, and models based on Bayesian networks. 

Recently, pushed by the need for continuous quantitative assessment of threats, which requires dynamic adaptation of defenses in networked systems, Bayesian networks have become adept as they allow reasoning about network states and the causal dependencies of state transitions. One of the most prominent approaches for dynamic risk management using Bayesian networks is the work of Poolsappasit et al. [4]. Their threat modelling approach combines asset identification, system vulnerability, and connectivity analysis, as well as mitigation strategies. The work focuses on the likelihood of attacks rather than other risk factors, such as the impact or costs. Xie et al. [5] also used Bayesian networks for security risk analysis of networked systems relying on runtime observations from intrusion detection systems to evaluate security risks. Dantu et al.’s [6] approach to security risk management also relies on Bayesian networks that capture the influence of the attacker profile on risk estimation. 

However, the full potential of Bayesian networks is realized when conditional probabilities of attack events are known together with preconditions and the order of network state transitions, which may be extremely challenging in the domain of distributed systems composed of multiple services and infrastructures as in IoT and cloud-based smart grids. In these scenarios, where the protection of some of the components falls beyond the responsibility of in-house engineers, the security conditions and the cause–consequence relationships become blurred. Furthermore, the tool support dedicated to the analysis of Bayesian networks for security is limited [2], which is not the case of attack tree-based methods.

In any case, both approach trends are not opposite to one another, but in fact, they can converge as demonstrated by Qin and Lee [7], who proposed the conversion of an attack tree to a Bayesian network by adding an order to actions in the attack tree nodes connected by AND logic gates.

Therefore, this paper proposes the use of the attack defense trees (ADTs) introduced by Kordy et al. [8] for reasoning on the initial estimation of system risks in multinetwork scenarios, where limited or no information is usually available about the order of the attack events or about the possible effects that some attack events may have on others, since they may target different parts of the composed system. In this work, the refinement of such estimation during system operation is also supported by ADTs, which could be enhanced in the future with conditional probabilities learned from continuous monitoring of the system and threat intelligence inputs.

Since being originally proposed by Schneier [9], attack trees (ATs) have been extensively studied as an easy-to-understand, reusable, and effective formalism to analyze security threats by focusing on how potential attackers may try to attack systems. Attacks against a system are modelled in an acyclic tree structure, where the root node represents the attack main goal, the branches in the tree represent the different paths an attacker can follow to achieve the main goal, and leaf nodes represent elementary attack events. Branches in the tree are formed by logic OR gates that represent alternative ways to fulfil a goal, and AND gates that model conjunctive sub-goals, which all need to be fulfilled for the attack to be successful. 

Attack defense trees (ADTs) [8] and attack countermeasure trees (ACTs) [10] extend the AT concept by adding to the attack model information on possible defenses or countermeasures that the defenders of the system may adopt to try to prevent the success of the attack. While being very similar, the main difference between ACTs and ADTs is that in ACTs, the countermeasures can be placed at any attack node of the tree, and in ADTs, only at attack events (leaf nodes). Furthermore, the ADT model and the reasoning it supports have been thoroughly formalized in [8] and [11]. 

In order to aid in the quantitative evaluation of how attack and defense parameters may impact on the main attack goal, ADTs can be enriched by attribute decoration to both attack and defense nodes with different techniques, such as Amoroso [12], Mauw and Oostdijk [13], Buldas et al. [14], Edge et al. [15], and Wang et al. [16]. These analyses are usually aimed at quantitative reasoning on different attack-defense scenarios towards informed decision-making of countermeasures [10,17]. ADTs have been shown to be efficient in this purpose in deep studies, such as those offered by Byres et al. [18] on SCADAsystems, Henniger et al. [19] on vehicle communication systems, and Abdulla et al. [20] on the GSM radio network to name a few. 

All these previous works considered individual attribute domains and some of them studied the derived attributes like risk as well: Salter et al. [21] proposed probability, impact, cost, severity, skill level, and consequence; Edge et al. [15] first introduced defense cost; Byres et al. [18] introduced the detectability of the attack, difficulty or skill level, and attack time; Buldas et al. [14] and later Jürgenson and Willemson [22] proposed different methods to compute the expected outcome and expected penalty of the attacker; Fung et al. [23] proposed a metric of the difficulty level to compute the scenario survivability; and Roy et al. [10] studied the defense cost, attack impact, risk, and return on investment in attack defense trees. An extensive survey of the quantitative attributes in ATs and ADTs can be found in [11]. Still, none of them have studied the quantitative analysis of ADTs with risk as the fundamental attribute being considered. As it will be shown, in our methodology, we propose an algorithm to compute the risk vector in each of the ADT tree nodes and conclude that it is necessary to evaluate first both individual attributes (probability, impact, cost) and the derived risk attribute in the leaf nodes to propagate the risk attributes to ascendant nodes. 

Following the line initiated by Rios et al. [24] on continuous security risk assessment, this work advances the risk evaluation techniques for distributed systems by making use of attack and defense risk attribute quantification in ADTs. The methodology addresses the needs of systems that orchestrate multiple internal and third-party components, where it is necessary to take into account the role of services or components whose protection lies outside of the hands of the smart grid operator. In addition, the methodology includes refinements of the risks assessed in the system design once the actual external services are selected for outsourced components and further refinements during operation through continuous monitoring of the countermeasures deployed to control risks. Please note that our methodology fits within step 2 “conduct assessment” of National Institute of Standards and Technology (NIST)’s Special Publication SP 800-30 Guide for Conducting Risk Assessments [25], offering a solution to quantitative computation of the overall system risk.

## 3. Quantitative Risk Assessment Methodology

Our approach to the quantitative risk assessment in smart grid systems promotes early updates in risks evaluated at the development time by assessing at all times the security status of the system in operation. The methodology integrates into the calculation of the risks the influences of the deployment of the different system components in the chosen infrastructures (e.g., in the cloud), as well as the current status of the defenses and attacks in operation, thus continuous risk assessment is offered. In order to put the focus on the risk assessment techniques used, details of the performance of continuous monitoring were left out of the scope of this paper, which can be consulted in [24], although a summary of the approach and how it makes the continuous refinement of risk status possible is explained in Section 3.5. 

Continuous risk management involves the identification and initial evaluation of the risks over system assets followed by the continuous monitoring of the evolution of the risk severity level. This implies the continuous assessment during system operation of the status of the risk attributes identified during the design phase so the risk level can be tuned according to the actual occurrence of attacks or their symptoms, as well as the status of deployed defenses.

Figure 1 shows the iterative process proposed to systematically perform continuous risk management in smart grids. The overall process consists of five main steps, namely:System ADT modelling, where the system security analysts create the system ADT representing the potential attack-defense scenarios.Risk assessment over system ADT, which consists of the evaluation of system risks by setting the risk attribute values to the tree leaf nodes representing the possible attacks and required defenses, and propagating the values up to the top tree node.Risk sensitivity analysis over system ADT, where the effects that different attribute values of both attacks and defenses have on the system risk are studied.Risk-based optimization of defenses to select system protections according to different combinations of constraints, such as minimizing system risks with a limited security budget.Continuous refinement of risk evaluation in the system operation phase of the status of attacks and defenses in operation, which enables early feedback to risk assessment and continuous tracking of the system risk situation.

As shown in Figure 1, two major risk refinement loops are considered in the methodology. The first risk assessment rectification occurs after service providers are selected for third-party components and the offered defenses are known. This refinement is made once the system components’ deployment options are decided, and deployment is made. Subsequent iterative refinements occur when the system components are in operation. The continuous monitoring of the performance of security defenses in the components will offer information on the status of potential attacks on the components and of their countermeasures, which allows for refinement to more realistic values regarding the risk parameters initially estimated in system risks, such as the probability of a defense failing, the protection effectiveness to reduce attack impact, etc. In the following sections, the details of each of the methodology steps are provided.

### 3.1. Step 1: System ADT Modelling

The first step of the methodology is the modelling of system attack defense trees, which consists in creating the ADT models capturing the potential attack scenarios against the system as well as the respective defensive controls that may be adopted to counter the attacks. In this section, we explain the proposed approach to build the ADTs that represent diverse attack scenarios to the system and how they can be integrated into a single system ADT, which enables the evaluation of overall system risks. In the methodology, the idiosyncrasies of complex composite systems are addressed by deriving from the system ADT the set of attack events and controls that correspond to each of the system assets, so later, the risk analysis on particular assets or components is possible.

Following a hierarchical attack modelling approach, for each attack-defense scenario envisaged, an ADT is created, where the high-level potential threat is represented by the root node, which is decomposed into lower-level threats represented by intermediate nodes. The tree leaves are the attacker actions, which exploit particular vulnerabilities of the system assets and therefore are not further decomposable. In general, attack actions against the system assets (components) depend on the component nature, type, interfaces, etc. Defenses or protection that system developers may adopt to counteract the external attack actions are represented as being associated to the attack events in the lower level. Figure 2 depicts the ADT structure (modelled with the ADToolRisk tool described in Section 4), where attacks are represented by ellipses in red and countermeasures (defenses) by rectangles in green. Attack goal refinement relations are drawn as solid edges between nodes, while defenses are connected to the countered attacks by dotted edges. Two types of refinements from parent to children nodes (all of the same type) are possible: (i) Conjunctive refinement (AND) depicted by an arc, which connects the parent’s edges to its children; and (ii) disjunctive refinement (OR) with no mark in the graph.

As it can be seen in Figure 2, the tree structure of ADTs facilitates reasoning on whether the attack sub-goals collectively (conjunctive sub-goals joined by the AND operator in the parent) contribute to their parent goal achievement or alternatively (disjunctive sub-goals joined by the OR operator in the parent). Similarly, ADTs illustrate whether the defenses contribute jointly to the parent countermeasure mechanism (joined by the AND gate in the parent) or are alternative solutions (joined by the OR gate in the parent). This will allow for quantitative expression of both: (i) Attack events’ contribution to the system risk severity level, and (ii) defensive controls’ contribution to threat mitigation and therefore to risk severity level reduction.

Adding defenses to individual attack events is a non-trivial task that requires expertise in security and privacy mechanisms for the system architecture components under study. In smart grid scenarios that are based on IoT technology, knowledge of cloud security issues and adoptable measures is needed, since IoT can be supported by cloud resources at the application layer [26,27]. The MUSA security metric catalogue [28] is a comprehensive collection of threats and security controls that can aid in this task. The threats in the catalogue are mapped to both controls that could be used to counteract the threats and metrics over the controls, which helps in evaluating the control performance. The controls in the catalogue follow the NIST SP 800-53 taxonomy [29], which facilitates the auditability of defenses in the ADTs as these controls are mapped to other standards, such as Cloud Control Matrix [30] and ISO/IEC 27001 [31].

When modelling defenses in the ADT, system internal components need to be self-assessed first in order to know which protections are already implemented, so as to discard a number of potential damages. In these cases, the potential attack should be represented as a tree node together with the defense implemented in the asset. Security self-assessment techniques that can help in identifying appropriate defenses to be modelled in ADTs were studied by OWASP [32], Berkley [33], and CSA [34].

Envisaged attacks on outsourced components also need to be modelled in the ADT together with the defenses that would be requested of the external providers. This issue is extremely important in modern smart grids since they are distributed systems, where multiple outsourced components may exist. The selection of system defenses will need to consider those available from the external services that the system will use (e.g., infrastructure-as-a-service, usually private; software-as-a-service; IoT edge services; etc.) and the affordable security expenses in third-party components. 

In composed applications, where the overall system is constituted by multiple collaborating parts or components, a single ADT for the complete system, namely the system ADT, is created, representing, in a single tree node structure, all individual attacks and defenses to be deployed in system components. As a result, the overall system risk level is computed on top of the system ADT created and all paths and relationships between nodes are taken into account when calculating risk metrics according to different node configurations, as explained in Section 3.2. 

For large composite systems or systems where many individual ADTs have been modelled, building a unified system ADT may lead to a too-large tree structure whose visualization is no longer easy, and therefore, it is advisable that the set of individual ADTs is maintained together with a simplified system ADT, where its root node has the root nodes of all individual disjunctive ADTs as children nodes aggregated by an OR relationship between them. 

### 3.2. Step 2: Risk Assessment over System ADT 

The quantitative risk analysis starts by decorating the attack events and the defenses in the system ADT with estimated values for the proposed risk attributes as explained in the rest of the section. Once the attribute values of the leaf nodes are defined, different metrics on the ADT can be obtained by propagating these values up to the tree root node. 

#### 3.2.1. Attack Risk Attributes Decoration

Multiple information security risk evaluation methods exist in the literature [35], most of which are based on calculating the risk level by multiplying the *probability* of the attack to be successfully realized by the *impact* the attack would have over the system (i.e., the damage or penalty to the system). This assessment has a very extended variant, which considers the *cost* of the perpetration of the attack to explicitly capture the idea of the higher the effort level for the attacker (i.e., the amount of resources required by the attacker), the lower the risk of the attack. 

In our methodology, we propose the use of the three-attribute-based risk evaluation in Equation (1), as it facilitates the analysis of the cost-effectiveness of the defense strategies adopted in order to minimize system risks:(1)$$R_{i}=\frac{P_{i}\times I_{i}}{C_{i}},$$
where i represents each threat or potential attack in a set of T threats against the system, i.e., i∈[0, T].

With the values of these three attributes together with the resulting risk value, a risk attribute vector $\{P_{i},\;I_{i},C_{i},R_{i}\}$ is built, where $P_{i}$ is the *probability* of the success of attack, $I_{i}$ is the *impact* of the attack on the system, $C_{i}$ is the *cost* of the attack, and $R_{i}$ is the *risk severity* evaluated by using Equation (1).

It is important to note the units and potential value ranges of the operands in the formulas so the resulting risk level is meaningful. The successful occurrence probability values fall in the [0,1] interval, while the impact values are usually between 0 and 10, with 0 expressing no impact and 10 the maximum impact over the system. Threats with 0 likelihood or 0 impact are not worthy of consideration for risks, so the lowest limits are usually not used in the standard guidelines (e.g., ISO/IEC 27005 [36] and OWASP Risk Rating Methodology [37]). 

The units and values for costs in Equation (1) can be given in dollars, euros, man-hours, or generic cost units, which later need to be converted into money units using a previously selected conversion factor. Considering that attack costs can range from 0 money units to infinity, the risks values calculated by making use of Equation (1) may easily lead to apparently negligible risk values in the order of thousandths of severity or lower. For this reason, we propose the adoption of a normalized scale for attack costs and defense costs, where the costs are normalized to the node with the minimum cost in the tree. 

In order to avoid issues in the risk computation of Equation (1), it is advisable that nodes with zero costs are decorated with an approximation of $10^{-n}$ cost units, where n is a natural value. In this way, the cost scale will be within the [$10^{-n}$, 10] interval, which renders risks of the [$10^{-n-1}$, 10] interval, where low n values make it easier for the analysts to compare risks between tree nodes.

In cases when third-party software components are part of the composite system architecture, such as in smart grid settings where data storage or processing cloud services are outsourced, the modelling of their attacks and attribute decoration shall be done in the system ADT with the information on potential attacks obtained from the provider if possible, or from a prior study on potential threats against the component from previous surveys or experiences with components of similar characteristics. 

#### 3.2.2. Defense Risk Mitigation Attributes Decoration

Defenses or countermeasures can also present risk attributes related to their cost-efficiency in mitigating threat risks. To model this, we propose decoration of the defenses in the system ADT with three risk attributes similar to those proposed for threats: (i) The *probability* of the defense to successfully counteract the attack event, which ranges in the interval [0,1]; (ii) the *impact* the defense can protect as a percentage of the impact of the attack that can be avoided when adopting the defense, whose interval is [0,10]; and (iii) the normalized *cost* of the application of the countermeasure mechanism for the defender, which ranges within the [0,10] interval as well. Equation (1) is used for defense *risk mitigation cost-efficiency level* evaluation and the risk vector for the defenses includes the four attributes $\{P_{i},\;C_{i},\;I_{i},\;R_{i}\}\;$.

The defense decoration for outsourced components shall be done from the safeguards specified in the security service level agreement (SLA) offered by the provider, which indicates the controls implemented together with their price. Usually, even if the third-party SLA includes the available controls, the corresponding threats or attacks are not explicitly indicated, and system security experts should add in the ADT attacks associated to each of the defenses in the SLA in case they were not yet identified. 

#### 3.2.3. Estimation of Risk Attributes on Leaf-Nodes

One of the most important steps in the methodology is the initial estimation of the risk attributes’ values for both attack events (leaf nodes in the attack tree) and for the adoptable defenses to mitigate their impact. The attribute value assignment is usually performed by security experts based on their previous experience and according to the assets’ and system characteristics. Knowledge on hacking tactics as well as attack detection and analysis by system operators and experts will definitively help in risk values’ assignment. Countermeasure attribute value estimation will require the collaboration of both designers and operators as defensive or reactive protection could be adopted in the system with diverse costs and effectiveness. Different techniques for ADT decoration were studied by Bagnato et al. [38] and [39], which may serve as a reference.

The *probability* of an attack being successful and its *impact* value can be estimated as aggregations of multiple factors. For example, the OWASP methodology [37] proposes the consideration of eight *likelihood* or *probability* factors grouped in two main categories: Four factors related to the vulnerability exploitability and four related to the threat agent capacity to exploit the vulnerability. Similarly, the OWASP method also considers eight factors to assess attack *impact*, including both technical and business aspects. Other standards, such as ISO/IEC 27005 [36] and ISO/IEC 29134 [40], also provide guidelines on how to estimate the likelihood and impact of security and privacy threats, respectively.

In smart grid systems, where one or multiple services are consumed from third-party service providers, such as virtual machines in private clouds that host system components, modelled attack scenarios will likely include attack events to those outsourced assets as well as potential countermeasures to counter them. In these cases, it is necessary to perform an initial estimation of the risk attributes of those attack events and defenses too.

The continuous monitoring during operation of the status of the system services, their defenses, and attack symptoms will allow the risk attributes’ vectors $\left\{P_{i},\;C_{i},\;I_{i},\;R_{i}\right\}$ in the attack event nodes and in their respective defenses to be refined, as explained later.

#### 3.2.4. Risk Assessment in Countered Nodes

When adding a defense to a leaf attack in the ADT, the risk posed to the system by the countered attack is modified by the risk mitigation effectiveness of the defenses and vice versa, when an attack event is modelled as targeting a defense in the ADT, and its protection effectiveness is weakened. Therefore, defenses in ADT act as countermeasures to attacks, and conversely, attacks act as countermeasures of defenses. Therefore, if we generalize this situation for both the proponent (attacker) and opponent (defender) perspectives of the ADT, a method for calculating the risk attributes in countered nodes is necessary. Table 1 presents the rules proposed to evaluate the nodes countered by nodes of the opposite type.

As expressed in Table 1, the success of a countered node is reduced when its safeguarding opposite node succeeds. Therefore, the success probability of a countered node can be computed as the success probability of the node multiplied by the probability of the countermeasure failing (i.e., 1 minus the success probability of the countermeasure).

The impact on the system of the countered node is lowered by the impact of the countering node because they have opposite directions in the protection of the system. That is, the damage caused on the system by an attack is reduced if a corresponding defense is implemented in the system. Similarly, when an attack is designed against a defense, it reduces the defense system’s protection effectiveness.

Finally, the cost of a countered node is not affected by the cost of the countermeasure because in the worst-case assumption, the proponent has no information about the costs of the actions by the opponent. In fact, while the attacker’s costs relate to the means they use for attacking the system (e.g., exploiting software, botnet node infrastructure, etc.), the costs for the defender include all resources employed in the system’s protection (antiviruses, purchased security services in the cloud, security hardware, detection system infrastructure, developer costs, etc.). A priori, these costs are independent, as the attacker will launch the attack campaign with all the available resources, being ignorant of whether the defender has invested in implementing defenses against them. 

#### 3.2.5. Risk Propagation Algorithm

In order to quantitatively evaluate the risk of the system to be attacked, it is necessary to calculate the risk attribute vector $\left\{P_{i},\;C_{i},\;I_{i},\;R_{i}\right\}$ of the root node “attacking the system” in the system ADT. To this aim, a bottom-up propagation algorithm, which propagates the risk vector from leaf nodes up the logic tree hierarchy, is needed. There are several approaches in the literature for attack tree and ADT attributes’ propagation rules to compute the utility value of the tree root node, such as those proposed by Weiss [3], Buldas et al. [41], and Edge et al. [17]. 

In our approach, we adopt the principles of Weiss [3] by assuming the worst-case scenario, where a smart adversary would intelligently apply all available resources to attack the system. This assumption influences the risk bottom-up propagation rules in the ADT as long as the behaviors of the AND operand and the OR operand differ in the evaluation of risk attributes from their children as follows. 

The risk associated with an AND node is calculated in terms of the sum of the efforts of the children. That is, while the satisfiability (probability of success) of the parent requires that all the children are satisfied, the cost for the parent is the sum of the costs of the children nodes, and the parent impact also aggregates the children impacts. In the impact case, the formula proposed is that of Edge et al. [17], which accommodates the fact that in most cases, the effect over a system of a set of successful actions is greater than the sum of the individual events. The risk of a parent OR node is the maximum of the risks associated with its descendants as the smart adversary will choose to carry out the attack that has a higher probability of success and produces the highest damage with respect to the expenditures of performing the attack. Therefore, the OR operand requires a local optimization with respect to risk.

In summary, the proposed smart adversary case’s risk assessment over ADT derives the risk attribute vector of the root node in the ADT as a result of the bottom-up propagation of the risk vectors of child nodes to parent nodes by the rules defined in Table 2. 

As in Weiss’ proposal [3], the specificity of the risk propagation algorithm proposed resides in the need to first compute the individual attributes (probability, impact, cost) and the derived attribute (risk) for all the children in order to obtain the values for the parent node. An advantage of our method with respect to the Weiss method is that instead of relying on the empirical assessment of impacts, we adopt Edge et al.’s [15] formulation for this attribute, which enables risk estimations prior to system deployment.

In OR cases, where two or more children have the same risk Ri, the node with $max_{Ri}$ among them is selected as the one with the highest probability value, or the one with the highest impact value if both have the same probability. In the case where all the children have the same risk vectors, the OR parent adopts the risk vector of the first child.

As can be seen in Table 2, our approach leads to commutativity and associativity of the ADTs, while the ADT distributivity is not guaranteed. Therefore, in our method, the calculated risk vector for the root node in the attack defense tree T1 = A ∨ (B ∧ C) is not necessarily equal to the risk vector evaluated for the equivalent binary formula, T2 = (A ∨ B) ∧ (A ∨ C), where A, B, C, T1, and T2 are all attack defense trees, and T1 and T2 are semantically equivalent. This is because in our method, the worst-case assumption implies that in disjunctive options (OR operands), which are not equally equivalent in risk weight, the decision will be made on the one with the highest risk weight. Hence, in cases where A has an intermediate risk value between B and C, the result of T1 = B ∧ C is not equal to T2 = B ∧ A for ordered risk values B > A > C or T2 = A ∧ C for C > A > B.

As a result, as described by Jürgenson and Willemson in [22], similarly to other risk propagation methods that propose local maximums, such as Weiss [3], Buldas et al. [41], and Edge et al. [17], our risk vector propagation method only has partial consistency with the semantics framework by Mauw and Oostdijk [13], who established the foundations for attack tree semantics and advocated for the commutativity, associativity, and distributivity of the conjunctive combinator (AND gate) and disjunctive combinator (OR gate) of attack trees so attack trees can be transformed to logically equivalent attack trees. However, our method reflects a more realistic paradigm, where only some of the candidate options represent the same risk to the system and therefore a smart adversary would not have the same appetence for all of them.

It is interesting to note that our methodology addresses both perspectives of ADT, the proponent (attacker) perspective and opponent (defender) perspective, and the computation rules for risk attributes are the same for both, which makes our method adhere to the attack defense tree foundations proposed by Kordy et al. [2].

#### 3.2.6. Risk Severity Metrics Proposed

Once the risk vectors are estimated for the attack events in the ADT, the risk landscape for the system can be obtained by depicting all the attack event risks in a two- (probability, impact) or three-dimensional (probability, impact, cost) space. For simplicity and similarity with the guidelines by standard risk frameworks like ISO/IEC 27005 and OWASP, the two-dimensional space depicted in Figure 3 is usually preferred. However, it should be noted that cost values can be considered within the probability attribute. Then, the attacks can be categorized into buckets of high, medium, and low risk, and ranked by their severity and position in the quadrants as in the OWASP risk rating methodology. The defense strategy thus starts fixing vulnerabilities associated with attacks with the highest scores. 

When risk quadrants are used, relevant metrics related to the density and risk severity of the threats within the critical quadrant, such as the ones proposed in Table 3, should be studied, because these are, a priori, the most problematic risks for the system. The minimum and maximum values for both the probability and impact of threats within the critical quadrant reveal the points that limit the interval that should be studied. While the threat density indicates the amount of critical threats to be considered, the risk center of mass indicates the mean probability and mean impact in the critical quadrant. The mean risk value in the critical quadrant is calculated using the two metrics in Equation (1). The threats showing the maximum risk and minimum risk in the critical quadrant are the ones in the region with the highest and lowest values in Equation (1), respectively. 

Please note that the metrics proposed in Table 3 should be used only with sets of independent threats whose comparison makes sense, that is, the set of elementary threats in the leaf nodes or the set of attack scenarios represented by the root nodes of disjunctive individual ADTs that compose the system ADT.

Being complementary to the traditional approaches that consider attacks (and their respective controls) as independent events, we propose an evaluation of the overall risk of the system by considering attack events’ relationships defined by the ADT, which enables risk-driven design of protection strategies based on the outcomes of the risk sensitivity analyses explained below.

### 3.3. Step 3: Risk Sensitivity Analysis over System ADT

Risk sensitivity analysis over the system ADT enables informed decisions on system protection strategies. Provided that all the risk attributes of the leaf nodes in the ADT are estimated, the risk sensitivity analysis investigates which attack events and which defenses have a greater impact on the overall risk of the system. Moreover, the sensitivity analysis allows study of the risk variability in any of the tree nodes based on variations in the risk attributes of other nodes. 

From the ADT proponent or adversary point of view, studying the swings in attack event attributes permits identification of which attacks produce a higher risk at the system level, which are the minimum attack sets that realize the ADT at the lowest prize for the attacker, which do harm the system more, etc.

From the defender perspective, fluctuations in the defense attributes allow the deduction of which protection strategies minimize the overall risk or the overall attack impact the most, which is the cost to fully minimize the risk of the system attack’s success, etc.

Three main types of risk sensitivity analysis can be performed on ADTs:Risk sensitivity of attack attributes: This analysis studies the impact of the fluctuations in the value of the desired attributes (attack probability, impact, or cost) of one or multiple attack events (leaf nodes in the ADT) on the risk of the system (the root node) or of any branch in the tree. The analysis is relevant to the study of risk variations in ADT nodes caused by different increasingly successful (or increasingly impactful or increasingly costly) combinations of attack events. These studies are usually performed when the details of one or some attack events are subject to uncertainty or when there is a wish to simulate small variations in the parameters of attack actions so as to better understand their influence on the risk of the complete attack-defense scenario.Risk sensitivity to defense attributes: Similarly, the impact of the variations of defense attributes (defense probability, minimized attack impact, or defense cost) in the overall system risk or in the risk of any subtree or of any attack event can be studied. This analysis is conducted, for example, to understand to what extent the cost-effectiveness of the defense impacts attack risk minimization. By selecting more than one defenses in the analysis, it is possible to study risk variations in the ADT nodes caused by increasingly successful (or increasingly effective or increasingly costly) combinations of defenses. These studies are usually part of the defense strategy decision process when the attributes of one or a set of defenses are analyzed to test their impact on the attack-defense risk scenario.Risk sensitivity to combined attack and defense attributes: This analysis combines the two previous ones, where the value of the desired attribute (probability, impact, or cost) of a combination of defenses and attack event nodes in the ADT is progressively increased in order to study the effects on the risk values in the remaining tree nodes. This type of risk analysis can be made, for example, when adjusting the design of the defenses while there are some ambiguities on the initially estimated attack attribute details.

### 3.4. Step 4: Risk-Based Defense Optimization

Different constraints may arise to drive the optimization of defenses, such as a limited security budget or the need to protect a particular asset in the system. In our methodology, we use ADT mincuts (or minimum cuts of ADT) for the reasoning on the best defense strategies to be applied over both the individual system components and the system as a whole. 

*Mincuts* are different attack-defense minimal sets that realize the main goal. An ADT with n leaf nodes can, at the most, have $\left(\begin{array}{c} n\\ 2 \end{array}\right)=\frac{n\;\left(n-1\right)}{2}$ mincuts. It is important to note that a defense present in an ADT mincut covers every attack event in the mincut [10].

Consequently, we define a three-dimensional relationship matrix $T=f\;\left(AT_{i},\;D_{j},\;A_{k}\right)$ that is derived from the ADT mincuts, where the rows represent attack events ($AT_{i}$), the columns represent the defenses ($D_{j}$), and the layers or pages are mapped to system assets ($A_{k}$) as shown in Figure 4. Therefore, the element $t_{ijk}$ in T is $t_{ijk}$ = 1 when, for asset $A_{k}$, $AT_{i}$ is an attack against $A_{k}$ and $D_{j}$ covers $AT_{i}$, or else $t_{ijk}$ = 0.

The T matrix above can be used to solve the single-objective defense optimization problem when searching for the minimum set of defenses that fully cover all (full cover problem) or some (partial cover problem) attack events in the ADT. Most interestingly, the T matrix enables multiple objective defense optimization problems to be solved, such as the dual objective optimization problem found, for example, when, due to a limited security budget, the aim is to identify the optimal minimum set that covers selected attack events in the ADT while minimizing the cost investment. This can be expressed as:(2)$$F_{1}=\underset{\forall \mathrm{OPT}\;\in 2^{J}}{\min}\sum_{j=1}^{J}\mathrm{Cost}\left(D_{j}\right),$$
where OPT stands for the optimal and represents the minimum suite of defenses $D_{j}$ that cover all the selected attack events in ADT.

A similar dual-objective optimization problem arises when trying to identify the minimum set of countermeasures OPT that cover all the selected attacks while maximizing the system risk reduction obtained by the incorporation of OPT defenses, with the objective function:(3)$$F_{2}=\underset{\forall \mathrm{OPT}\;\in 2^{J}}{\max}\Delta Risk_{goal\_\mathrm{OPT}}.$$

In terms of economic gains, a dual-objective optimization problem also arises when trying to maximize the return on investment (ROI) for the defender when implementing the defense suite ***OPT*** employed in covering the desired set of attacks. ROI can be defined as the gains from risk minimization achieved by the *OPT* defense set, which can be expressed by adapting Sonnenreich’s definition of the return on security investment [42], since the cumulative cost of applying the *OPT* defenses is known. This objective function is captured as:(4)$$F_{3}=\underset{\forall \mathrm{OPT}\;\in 2^{J}}{\max}ROI_{\mathrm{OPT}}.$$

Algorithm (1) below describes the final algorithm created to solve the problem of dual-objective optimization by searching for the optimal combination of defenses, which, besides covering a specified critical set of attacks (CTS), optimizes a second weight variable, such as the defense cost. To this aim, the algorithm takes as input the T matrix, the CTS vector of attacks to cover, the AS vector of assets studied, and a vector W with the weights or values that the countermeasures take for the second variable to be minimized.
**Algorithm 1.** Finding the optimal defense set OPT covering the critical threat set CTS with minimum weight1ADMind (T, AS, CTS, W) 2Matrix = T3Assets_vector = AS4Attacks_vector = CTS5weights_vector = W6isValidcombination = false7Compact MatrixT by removing the pages not in AS and the attack rows not in CTS8Matrix = compact (Matrix, Assets_vector, Attacks_vector)9Calculate valid defense sets and minimize cumulative weight10combinationsSize = 2^Matrix.columnSize11best = infinite12ALL = []13**for** index **in** combinationsSize **do**14  // Calculate defense combination15  combination = index.toBinary()16  // Calculate the cumulative weight of the combination17  cost = sum(combination × weights_vector)18  // Check if current combination covers every attack in CTS19  **if** (Matrix × combination = { 1 }) **then** isValidcombination = true20  // Update the best so far21  **if** (isValidcombination) **then**22    **if** (cost < best) **then** {best = cost, OPT = combination} **end if**23  **end if**24  // Add to list of combinations25  ALL[index] = combination26**end for**27Display OPT and ALL

The values for risk and ROI are not directly obtained from the defense decoration in the ADTs but need to be calculated as a result of the bottom-up propagation of risk vectors in the tree following the rules in Table 2. Hence, for F2 and F3 optimization, the use of the algorithm in Algorithm (1) needs to be combined with the simulation of the system risk sensitivity to defense attributes. For example, to calculate OPT in F2, first, all the possible OPT sets that cover all the attacks are found by using the routine in Algorithm (1). Second, the risk sensitivity simulation is used to compute the ADT root node risk vector in all the possible defense scenarios. Finally, the OPT defense scenarios are extracted from all the possible scenarios and ranked by the highest risk attribute reduction in the root node. 

The study of countermeasure optimization at the asset level could be used to identify which countermeasures are the most appropriate to cover all the attacks and minimize the risks in third-party components and this, together with the system-level defense optimization, enables informed selection by providers offering the resulting optimal countermeasure set. The results for the system-level optimal countermeasure set may significantly differ with the optimal set identified to protect a particular asset, and the system security and privacy experts should decide on whether to invest in the reduction of the risks of a particular component or in minimizing the risks at the system level, which seems, a priori, more reasonable though other factors, such as defense costs (in internal and outsourced components), implementation time, etc., that may impact the final decision.

In our method, defense optimization at asset the level is a particularization of the system defense optimization. As the T matrix of the system ADT indicates which attacks and defenses are associated with a particular asset or component, by selecting the page of the T matrix corresponding to the asset under study, defense optimization can be performed considering that the covered attack set includes only attacks against the selected asset, i.e., attack rows that have an entry value equal to 1 in the asset page. 

### 3.5. Step 5: Continuous Refinement of Risk Assessment through Continuous Monitoring

The initially assessed risk needs to be revisited continuously during system operation as attacks may occur against the system assets, or the defenses deployed may not work properly, leaving the assets unprotected. Therefore, it is necessary to continuously re-evaluate system risks to include the effects of detected attacks or to update the controls deployed according to their actual status.

When using the ADT-based method proposed, continuous risk assessment consists in iterative refinements of the risk evaluation based on updates performed on risk attributes of attack and defense nodes made pursuant to the sensed system status. Continuous security monitoring allows for revisiting of the probability and impact values of attacks and defenses in the ADT, while the estimated costs for the attacker and the defender will require less frequent updates. In fact, the defense costs are set to the actual expenditures in the final security resources employed.

Whenever the monitoring system detects an attack event, its probability of success is set to 1, which raises the risk of the root node in the ADT. Similarly, when the damage caused on the system by a detected attack is studied, the impact attribute of the event node can be updated, and it may be possible that this information helps in the refinement of impact values for similar attacks in the ADT as well. Once the reaction mechanism to counteract the detected attack is in place, the corresponding defense is added to the ADT model and a new risk assessment iteration is performed.

Defense monitoring aids in updating the defense node attributes as well. Risk exposure increases whenever a deployed protection strategy fails, and new calculation of the root node risk is made by considering its probability of success as zero. Thanks to continuous defense performance checks, empirical tests of defense success and effectiveness during operation can be used to tune either the estimated success probability or the attack impact reduction ratio for the defense nodes in the tree. 

Please refer to [24] for a comprehensive description of a monitoring solution example addressing continuous checking of the status of both attacks and defenses in multicomponent systems that exploit distributed cloud resources. 

In summary, the methodology proposes to combine continuous monitoring of the system together with continuous surveillance on potential attacks against the system to be able to keep the ADT up to date as much as possible through accurate refining of the leaf nodes’ risk attributes in the tree and consequently the root node risk vector.

## 4. Validation in a Smart Building Energy Efficiency System

Our continuous risk management methodology was validated in a real-world smart grid scenario involving an energy efficiency system for a smart building by Tecnalia. This scenario is a good representative of modern electricity smart grid systems that combine distributed internal and external components, including both cloud and IoT resources. The setting allows for a simplified illustration of the complete process followed and demonstration of the feasibility and benefits of the continuous risk evaluation carried out.

Figure 5 depicts the simplified architecture of the smart building energy efficiency system under study. The building control SCADA in the smart building collects multiple sensors’ data (temperature, lightning, humidity, and presence) coming from Internet of Things gateways (deployed in wireless IoT edge devices) placed in the rooms as well as all building control data from Programmable Logic Controllers or PLCs (consumption of electricity, water, air conditioning, etc.). 

The energy efficiency management service is composed of a central management (EEMgmt) component and an energy efficiency artificial intelligence service able to perform smart estimations on all energy data coming from the building and stored in the energy metrics database All these components are deployed in virtual machines (VMs) from private cloud providers. 

In order to be able to validate the risk methodology proposed herein, we required the development of two dedicated tools supporting it. 

First, we created the ADToolRisk software tool, which supports the ADT analysis and reasoning on attack and defense strategies. The tool significantly enhances the open-source software tool ADTool [43] and automates the computation of the risk vectors in all the tree nodes by applying the countering rules in Table 1 and the bottom-up propagation algorithm shown in Table 2. The new tool implements an enhanced graphical interface that aids in the visualization of the resulting risk attribute vectors in all the nodes. Moreover, it offers powerful risk sensitivity simulation capabilities with respect to different risk attributes in both attacks and defenses as required in Step 3 of the methodology.

Second, a dedicated defense optimizer, the ADMind tool, was programmed in JavaScript language to automate the computation of all the defense combinations covering the specified set of attacks through the execution of the Algorithm (1). Furthermore, JavaScript enabled the inputs to be easily entered and displayed the optimization results in a web page while saving in program installation.

### 4.1. Step 1: System ADT Modelling

The modelling of the ADT for the case study system started with the study of all envisaged attacks and available defenses for all system components depicted in Figure 5. A security self-assessment was performed on internal components, which enabled the defense mechanisms they offered against potential attacks to be ascertained. The cloud components’ defenses were studied by analyzing the candidate cloud service providers’ service level agreements.

Due to space limitations and in order to ease the risk management procedure demonstration, the explanation is based on an extract of the ADT created, as shown in Figure 6 and named “steal energy data”, which is an appropriate representative of the system ADT for this use case. In fact, this ADT is composed of three disjunctive ADTs, “steal in origin”, “steal in transit GW->SCADA”, and “steal in storage” joined by an OR relationship, which indicates three potential independent means to steal the energy data by exploiting system vulnerabilities either when data is being captured, transmitted between components, or stored in databases. Note that for simplification of the example and explanations, some sub-goal branches were removed, such as stealing the data directly from the sensors, or stealing the data in transmission between SCADA and energy efficiency management or between DB and the energy efficiency AI component.

The taxonomy used for the attack events was adopted from the MUSA catalogue [28] while the countermeasures in the tree were named after controls from the NIST SP 800-53 [29] standard, which collected 912 fine-grained security and privacy controls. It is important to note that for some defenses, even if the same control name was used, the actual mechanism implementing the control depended on the nature of the asset. For this reason, the defense “RA-5 vulnerability scanning” (from NIST) was refined to “RA-5 vulnerability scanning-1” for the IoT gateway component, to “RA-5 vulnerability scanning-2” for the database, and to “RA-5 vulnerability scanning-3” for the virtual machines.

### 4.2. Step 2: Risk Assessment over System ADT 

The decoration of the system ADT for the use case assigned initially estimated the probability, impact and normalized cost values to all modelled attacks and defenses. Figure 6 shows the “steal energy data” ADT with all estimated attack and defense attributes (in yellow nodes) and risk vectors computed for all the nodes. Just after the decoration of all risk attributes for the leaf nodes in the tree, the value for the assessed risk vectors following the rules shown in Table 1 and Table 2 was automatically obtained and visualized by the ADToolRisk. Note that the risk vector always contains four ordered values (probability impact cost risk). The updated and initial risk vectors for countered attack events are shown in Figure 6 in the row just below the name of the event and in the row beneath, respectively. 

Table 4 contains the initially estimated risk vectors for the attack events shown in the columns corresponding to “risk vector before counter”, and next Table 5 the initial defense risk vectors shown under the “defense risk vector” title. The table includes in the “asset” column the target asset for each of the attack events identified. The notation used in the tables for attack events and countermeasures is At_i_ with i ∈ [1, 11] and D_j_ with j ∈ [1, 12], respectively, both numbered from left to right in the ADT of Figure 6.

Next, Figure 7 depicts the attack events’ risks within severity quadrants for the risk attributes of Figure 6. 

By using the severity quadrant limits from the OWASP methodology [37], all the attack events with high probabilities from 0.6 to 1 and high impacts from 6 to 10 fall within the critical quadrant. As shown in Figure 7, initially, these attacks were {At6_Man_in_the_middle, At9_Exploit_Vuln_in_VM, At10_DB_Account_Hijacking, At11_Read_Data}, and the addition of countermeasures resulted in only At6 remaining in the critical quadrant. 

Thanks to the proposed risk vector propagation method, the probability of success, impact to the system, and overall attack cost for the “steal energy data” main goal were deduced. As shown in Figure 6, the resulting risk vector in the ADT root node was {0.07, 4.2, 3, 0.1}, where the value risk 0.1 gives a measure of the system risk exposure after the effectiveness of all the defenses was estimated.

The comparison of this vector with the risk vector {0.54, 9.5, 8, 0.64} evaluated in the root node without countermeasures (when all countermeasures probability was set to 0) provides the risk reduction achieved, as in Equation (5):(5)$$Risk\_minimised_{all\_defenses}=0.64-0.1=0.54.$$

A reduction of 0.54 points in risk means that, by applying all the defenses modeled in the system ADT, an 84.37% risk minimization can be achieved with respect to not implementing any.

### 4.3. Step 3: Risk Sensitivity Analysis over System ADT 

The ADToolRisk was utilized to conduct different risk sensitivity simulations on diverse weighting factors, including attack attributes, such as the probability of success, impact to the system, or attack perpetration cost, and defense attributes, such as the probability, minimized attack impact, and cost. 

Sensitivity to attack attributes:

From the results of the initial risk assessment performed, the attack At6_Man_in_the_middle seemed to be the most relevant for the system. However, the risk sensitivity analysis with respect to the At6 attack attributes proved this statement is false with respect to system risk minimization.

Figure 8 shows the results of the ADT root node risk vector values when increasing the probabilities of At6 in 20 steps while leaving the estimated values for the defenses unchanged. As can be seen, with all defenses applied, the At6 node probability’s variation has no impact on the ADT root node whose risk vector always adopts the values {0.07, 4.2, 3, 0.1}. Similarly, Figure 8 shows the null risk sensitivity of the ADT root with variations of At6′s impact from 0 to 10. A similar situation was obtained when changing the At6 cost values from 0 to 10. This is the consequence of the application of the algorithm in Table 2, which makes the root node adopt the risk vector of the branch of the node named “steal in origin” as it is always more risky than the one of the “steal in transit” node, independently of the values of the At6 risk vector.

Therefore, our risk method throws light on the fact that even if the At6 node risk is critical, regardless At6 is countered or not, and the overall risk of the system (ADT root) does not depend on the risk attributes of this attack. Therefore, it is not worthy to spend a security budget on solely defending against At6 solely as other defense strategies are more efficient as demonstrated below.

2.Sensitivity to defense attributes:

The simulation of all the defense combination scenarios showed that the methodology proposed for risk analysis enables interesting insights. In this simulation, the probabilities of all the countermeasures were alternatively set to 0 or to the estimated value (first value of the risk vector in the defense nodes in Figure 6), which allows the ADT risks vectors of the 212 = 4096 combinations to be learned. 

As a result of these simulations, the countermeasure set {RA-5_Vuln_scanning-3, SI-20_De-identification, SC-7_Boundary_Protection} renders the system risk vector with the minimum risk {0.07, 4.2, 3, 0.1}. The cost of this minimum risk defense set is 15 cost units and they safeguard against the {At9_Exploit_Vuln_in_VM, At11_Read_data, At4_Steal_in_transit_sensor->GW} attack set, which differs from the set of attacks initially falling within the critical quadrant, as said before.

The set {SI-20_De-identification, SC-7_Boundary_Protection} gives a risk vector of {0.11, 9.3, 8, 0.13} with a slightly higher risk (0.13) at a security cost of 13 cost units. However, the probability of the success of the attack scenario (0.11) as well as its impact (9.3) are much higher than with the minimum risk defense set. Furthermore, the defense SI-20_De-identification by itself produces a very low risk in the ADT goal {0.1, 6, 3, 0.2} with only 7 cost units, which reveals it is the one applied in isolation that gets the highest utility in system risk reduction.

Figure 9 below shows the ADT root node risk vector sensitivity to the SI-20_De-identification probability. As can be seen, there is an inflexion point when the SI-20 probability surpasses 0.6, where the risk vector values of the propagation algorithm of Table 2 make the root node risk vector adopt the values {0.07, 4.2, 3, 0.1} of the “steal in origin” node. Therefore, in the inflexion point, the risk vector of SI-20_De-indentification loses its impact on the system overall risk. 

### 4.4. Step 4: Risk-Driven Defense Optimization

Thanks to the ADT-based defense optimization enabled by our method, the optimal countermeasure set that covers all the attacks with the minimum security cost (i.e., fulfils F1 in Equation (2)) was found to be {AC-6_Least_privilege, SI-4_System_Monitoring, SI-20_De-identification, SC-7_Boundary_Protection} at a cost of 19 units. This combination resulted in a system risk vector of {0.11, 9.3, 8, 0.13}.

As a result of the combination of system ADT risk simulations and matrix analysis on the ADT mincuts, the optimal minimum defense set fulling F2 in Equation (3), i.e., minimizing the risk of the system, was identified to be {RA-5_Vuln_scanning-3, SI-20_De-identification, and SC-7_Boundary_Protection}, and therefore, these were the main defenses applied in the corresponding components of Figure 5. As shown in the ADT of Figure 6, while SI-20_De-identification protects from vulnerability in the DB component developed by the in-house development team, the remaining two defenses in the set were requested from external providers in charge of implementing them. First, RA-5.3_Vulnerability_scanning protects from the At9_Exploit_Vuln_in_VM in the VM service in the cloud, and second, the SC-7_Boundary_Protection is used for protecting communications in the IoT edge. 

Therefore, the search of the IoT service providers for the edge device required the analysis of functionality matching plus the identification of which ones offered the required SC-7_Boundary Protection defense in its communications. 

Similarly, from all the available candidate private cloud providers, first, a selection of those providers matching the operational and functional requirements (such as the location of data centers, high availability, etc.) was made. The security and privacy controls offered were mapped to the standard NIST SP 800-53 [29] notation so a match with the modeled defenses in the ADT was possible. Furthermore, the probabilities and prices of the defenses offered in the cloud services were refined to reflect the actual efficiency and prices from the private service level agreements.

In the system ADT excerpt of our example, the provider of the VM hosting the DB offered two quality levels for the vulnerability scanning service on the virtual machine at different prices: The standard service at no cost and a premium service with cost of € 2000 annually. This meant that the initially planned cost for the RA-5.3_Vulnerability_scanning was not accurate as we planned for this defense 2 cost units, which means a total of € 20,000. 

Therefore, the next step consisted in revisiting the system ADT and refining the modeled countermeasures for the outsourced components with the updated information about the available defenses in the chosen providers together with their costs and efficacy in reducing impact. In the example, the cost of the RA-5.3_Vulnerability_scanning defense was updated to 0.2 cost units in the tree and its probability of success was reduced from 0.6 to 0.3, emulating the limited accuracy of the free vulnerability scanner in finding vulnerabilities. A similar process was followed with all the cloud protections required in the complete ADT.

The risk attributes of other protection strategies for other components were updated as well after the security investments were done. Among them, the probability of success of SC-7_Boundary_Protection for the GW was increased from 0.3 to 0.8 to reflect that a new security monitoring tool was installed in the communications of the GW, so its new risk vector was {0.8, 7, 6, 0.93}. 

After the optimal set {RA-5_Vuln_scanning-3, SI-20_De-identification, SC-7_Boundary_Protection} was exclusively applied in the ADT of Figure 6 and corrected estimations in the RA-5_Vuln_scanning-3 and SC-7_Boundary_Protection defenses were performed, a new risk evaluation was conducted, which resulted in a new risk vector {0.05, 9.16, 7, 0.07} for the ADT root, reflecting a further reduction of the overall system risk. This result is the consequence of the updates made in the SC-7_Boundary_Protection probability attribute rather than corrections to RA-5_Vuln_scanning-3, which did not affect the root node risk vector at all. The increase from 0.3 to 0.8 in the probability of the SC-7_Boundary_Protection success increases the likelihood of the “steal in origin” goal being satisfied to 0.02, which now shows a risk vector of {0.02, 4.2, 3, 0.03}. Since the “steal in storage” node risk vector is {0.05, 9.16, 7, 0.07}, according to Table 2, the root node adopts the risk vector of “steal in storage” as its child node with the highest risk value in the OR relationship between the children of the root node.

Similar refinements and analyses were done with the complete updated ADT, which was used for further improvement of the initial risk sensitivity analyses performed and in the following continuous evaluation and refinement of attacks and the defenses status in all the branches of the smart building energy efficiency system’s ADT. 

### 4.5. Step 5: Continuous Refinement of Risk Assessment through Continuous Monitoring

The continuous monitoring of the system was performed by using the MUSA Security Assurance Platform described in [24], which enabled insights into different parts of the system even in the cloud. The tool required that multiple monitoring probes of the network, system (IoT edge), and application (Gateway software) levels were distributed. Measures taken by the distributed probes about the status of the components were retrieved to the backend to continuously inform on, among other things, whether the deployed protections were working properly. In addition, defense agents, such as access control agents, were deployed as part of the system, which communicated their status and events to the MUSA Security Assurance Platform. This allowed continuous adjustment of the actions of the agents and monitoring of the status of these defenses. 

In parallel, to improve the raw estimations of the attack events in the ADT, continuous education and surveillance of the latest news about attacks that occurred in similar systems was necessary, as well as on all events that appeared in information sharing systems, which may be useful for the system. 

In the evaluated time frame, the energy efficiency system protections worked properly and all of the agent status was correct at runtime. However, a denial of service attack event was detected by the tool when monitoring the network, which affected the VM asset hosting the energy metrics DB. In order to prevent further incidents, it was decided to upgrade the RA-5_Vulnerability_scanning_3 on the VM to the premium service model as well as activate or improve the efficacy of other defenses in the ADT, according to the insight enabled by our methodology on defenses’ relevance for risk mitigation.

## 5. Discussion

This paper proposes a comprehensive methodology that enables informed decisions on security protections for smart grid systems through the quantitative assessment of cyber risks and the cost-effective optimization of countermeasures that minimize system risks. The solution is based on attack defense trees that capture the attack-defense scenarios of the system and the computation of proposed risk attributes that enable reasoning of how individual components’ risks impact the risks of the overall system. The work presents defense optimization techniques upon different optimization constraints and scenarios where varying efficiency and investments on attacks and defenses are considered.

The methodology has proved to facilitate quantitative risk estimation in scenarios where conditional probabilities between attack events are difficult or not possible to estimate due to target assets that are distributed in different networks or even fall beyond the control of the operators and limited information is available. This situation is expected to be common in future smart grids that combine multiple devices and services, both developed in-house and outsourced to external providers, e.g., storage and processing as-a-service in private cloud model.

The methodology makes possible a rationalized defense optimization method, which enables the identification of the minimum optimal defense set that is able to cover all or a selected set of envisioned attack events, i.e., the full or partial cover of attack event problems. Partial cover of the attack event dilemma occurs when, due to different hampering factors (e.g., limited security budget, limited available security mechanisms, etc.), it is not possible for the system administrators to implement all the protections necessary to cover all the potential attacks but only a subset of them. The selection of the subset of attacks could be the result of a previous analysis where a critical vulnerability set in the system is identified and only protections against attacks exploiting those vulnerabilities will be implemented, i.e., defenses against the critical threat set (CTS). 

The combination of the simulation of attack-defense scenarios over the ADT with matrix analysis of ADT mincuts makes it possible to solve the multi-objective defense optimization problems. In these cases, the optimal minimum set of defenses needs to cover all threats or the identified CTS, while fulfilling at the same time other constraints, such as available security cost or minimization of risk.

Furthermore, the defense optimization methods proposed enable all or only a subset of threats to be covered, e.g., only those affecting a particular system asset, which enables informed decisions on how best to balance the risks at the component and system levels. This information is relevant when protecting internal components and when requesting security controls to external providers, such as cloud service providers and IoT service providers.

As part of the future research lines, we plan to explore possible extensions of the risk assessment methodology to consider serialization in attacks and in defense application. Bayesian networks derived from the system ADT could be used for this analysis through the assignment of attack execution order and defense application order to nodes in the ADT and machine learning methods to estimate conditional probabilities in the nodes. 

## 6. Conclusions

In this paper, we introduced a holistic continuous risk assessment methodology for smart grid systems based on the use of ADTs to capture the relationships between threats and defenses in distributed system parts. The methodology can also be adopted in other types of complex systems with multiple components, where the risk assessment requires study of the potential attacks against multiple dispersed components, which may be developed internally or outsourced to cloud and IoT providers.

The ADT semantics proposed adopt the smart proponent and smart opponent approaches, where a smart adversary always selects the attack option with the highest risk weight for the system, and similarly, the smart defender always chooses the defense with the highest cost-effectiveness in risk mitigation.

Most importantly, the methodology allows prospections on defense strategies by enabling the simulation of diverse scenarios, where the risk attributes of the ADT nodes are configured to desired values so as different combinations of attacks and defenses can be evaluated. Therefore, our method supports security and privacy analysts in reasoning on both privacy and security risks in diverse scenarios over ADTs of the system and aids in the decision-making process on the best defenses to mitigate them. Moreover, the methodology enables analysts to rank the envisaged attack-defense scenarios based on the obtained risk attribute in the tree root node.

Hence, the methodology enables risk sensitivity analyses to be presented to decision-makers to guide in the prioritization of security investments. These analyses allow fine-tuning of the initial estimations made on attack and defense risk attributes as well as assessment of the variability of assessed system risks all along the system operation. 

## Figures and Tables

**Figure 1 sensors-20-04404-f001:**
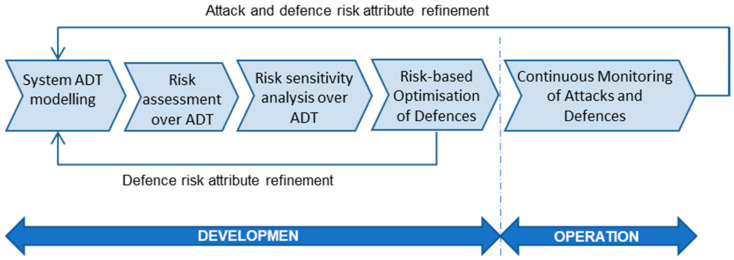
Attack Defense Tree-based continuous quantitative risk management process.

**Figure 2 sensors-20-04404-f002:**
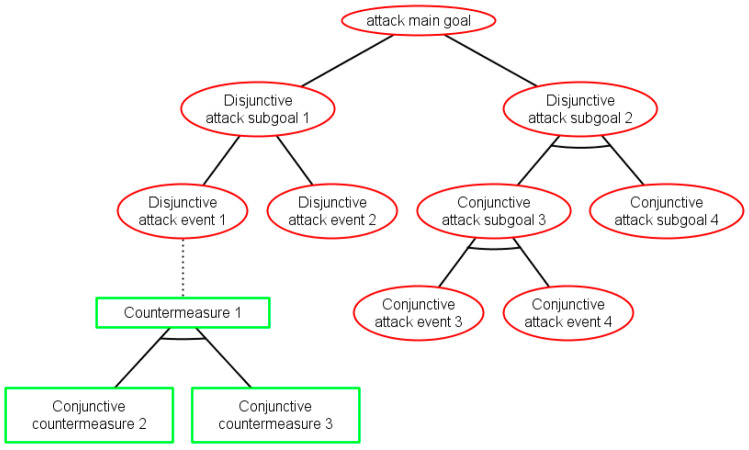
General structure of an attack defense tree (ADT).

**Figure 3 sensors-20-04404-f003:**
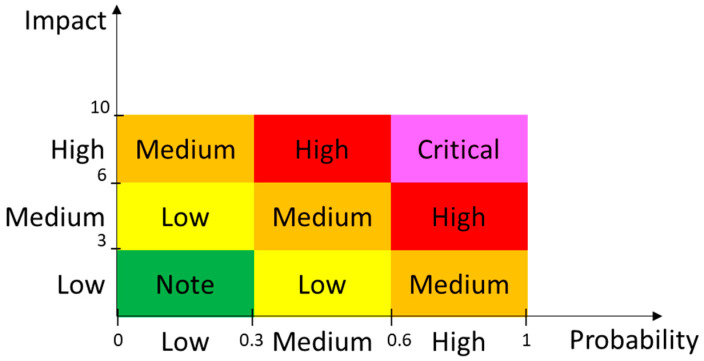
OWASP risk severity quadrants.

**Figure 4 sensors-20-04404-f004:**
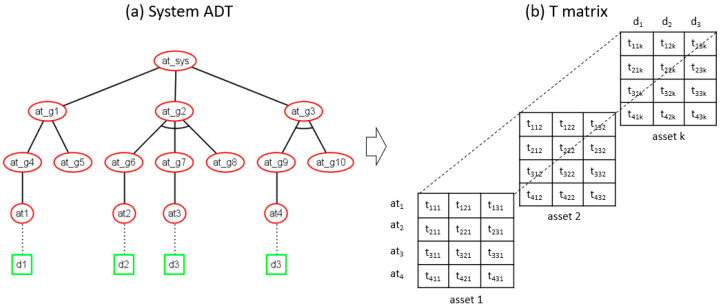
Example of a 3-D relationship matrix T (**b**) obtained from a system ADT with 4 attack events and 3 defenses on k assets (**a**).

**Figure 5 sensors-20-04404-f005:**
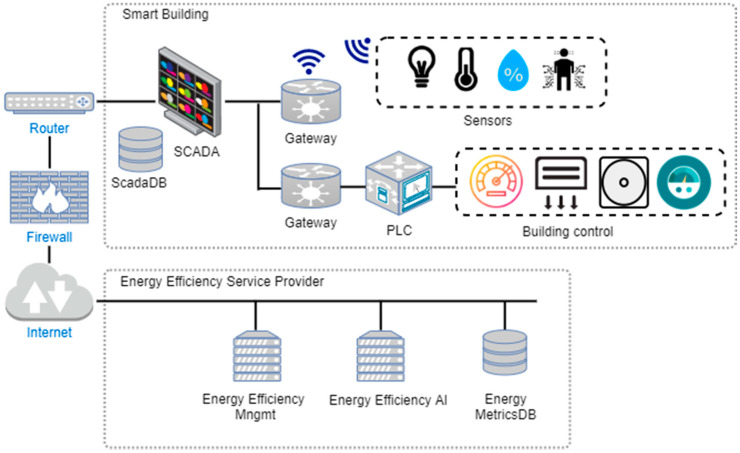
Simplified architecture of the smart building energy efficiency system under study.

**Figure 6 sensors-20-04404-f006:**
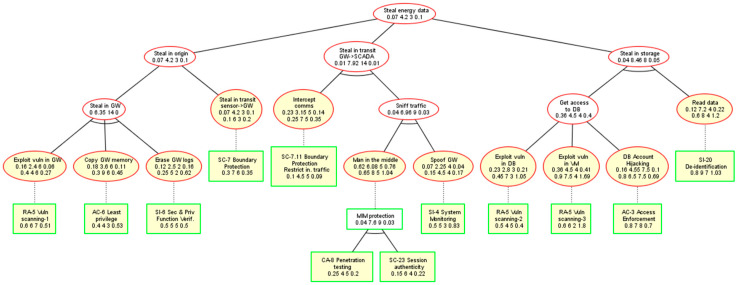
Use case ADT with risk vector evaluated in all the nodes.

**Figure 7 sensors-20-04404-f007:**
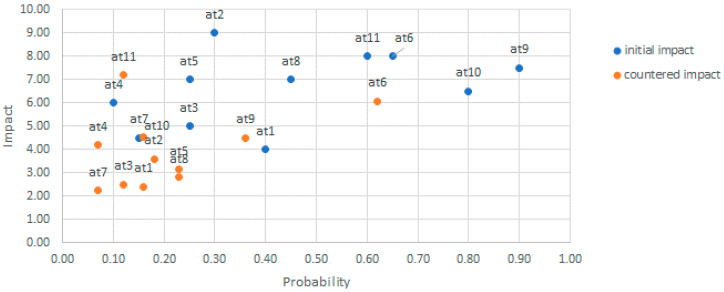
Severity of attack events in the use case before (blue) and after (orange) countermeasures.

**Figure 8 sensors-20-04404-f008:**
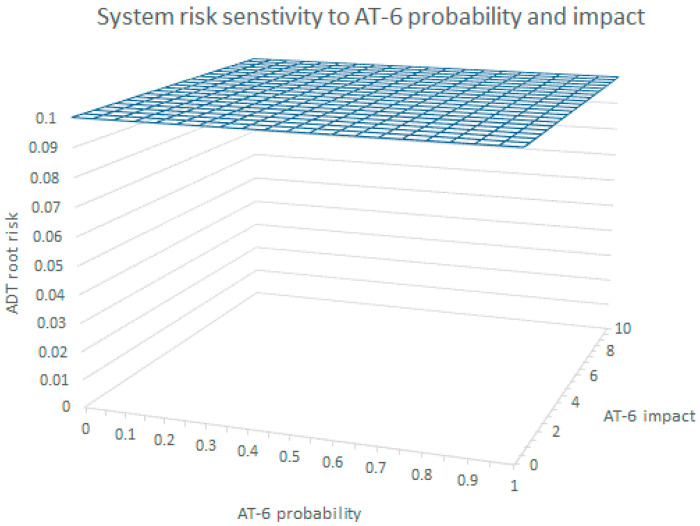
Risk sensitivity of the At6 attack event probability and impact.

**Figure 9 sensors-20-04404-f009:**
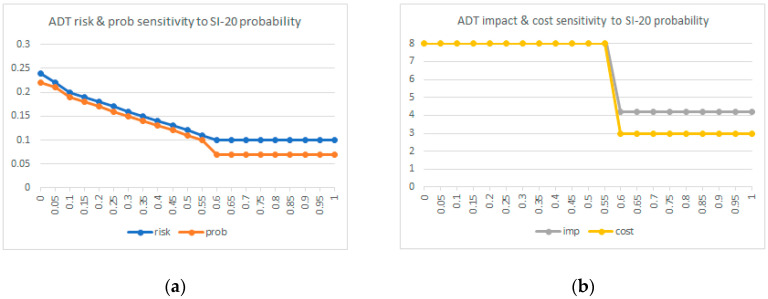
(**a**) Use case of the ADT root node probability and risk sensitivity to the SI-20 defense probability, and (**b**) use case of the ADT root node impact and cost sensitivity to the SI-20 defense probability.

**Table 1 sensors-20-04404-t001:** Risk vector evaluation rules for countered nodes in ADT.

Risk Attribute	Proponent	Opponent	Countered Proponent	Countered Opponent
Probability	$P_{p}$	$P_{o}$	$P=P_{p}\times \left(1-P_{o}\right)$	$P=P_{o}\times \left(1-P_{p}\right)$
Impact	$I_{p}$	$I_{o}$	$I=I_{p}\times I_{o}/10$	$I=I_{p}\times I_{o}/10$
Cost	$C_{p}$	$C_{o}$	$C=C_{p}$	$C=C_{o}$
Risk	$R_{p}=P_{p}\times I_{p}/C_{p}\;$	$R_{o}=P_{o}\times I_{o}/C_{o}$	R=P×I/C	R=P×I/C

**Table 2 sensors-20-04404-t002:** Risk vector propagation rules in ADT for Equation (1).

Risk Attribute	AND	OR
Probability ^1^	$P=\prod_{i=1}^{N}P_{i}$	$P=P_{maxRi}$
Impact ^1^	$I=\frac{10^{N}-\prod_{1}^{N}\left(10-I_{i}\right)}{10^{N-1}}$	$I=I_{maxRi}$
Cost ^1^	$C=\sum_{i=1}^{N}C_{i}$	$C=C_{maxRi}$
Risk	R=P×I/C	R=P×I/C

^1^ N: number of children in the gate, maxRi: the child with maximum risk computed with Equation (1).

**Table 3 sensors-20-04404-t003:** Risk level severity metrics proposed.

Risk Metric Name	Risk Metric Value
Threat density in the critical quadrant	*T* = total number of threats in the critical quadrant
Point of maximum risk in the critical quadrant	$P_{max}=\underset{T}{\max}P_{t}$ $\;,\;I_{max}=\underset{T}{\max}I_{t}\;$
Point of minimum risk in the critical quadrant	$P_{min}=\underset{T}{\min}P_{t}$ $,\;I_{min}=\underset{T}{\min}I_{t}$
Risk center of mass in the critical quadrant	$\bar{P_{T}}=\frac{1}{T}\sum_{t=1}^{T}P_{t}\;,\;\bar{I_{T}}=\frac{1}{T}\sum_{t=1}^{T}I_{t}\;$
Maximum risk in the critical quadrant	$R_{max}=\underset{T}{\max}\left(P_{t}\times I_{t}\right)$
Minimum risk in the critical quadrant	$R_{min}=\underset{T}{\min}\left(P_{t}\times I_{t}\right)$

**Table 4 sensors-20-04404-t004:** Risk vectors for attack event nodes (before and after countermeasures) in the steal energy data ADT.

Attack Event Id	ADT Node Name	Risk Vector before Counter				Risk Vector after Counter				Asset
		Prob	Imp	Cost	Risk	Prob	Imp	Cost	Risk	
At1	Exploit Vuln in GW	0.4	4	6	0.27	0.16	2.4	6	0.06	GW
At2	Copy GW Memory	0.3	9	6	0.45	0.18	3.6	6	0.11	GW
At3	Erase GW logs	0.25	5	2	0.63	0.12	2.5	2	0.16	GW
At4	Steal in transit Sensor -> GW	0.1	6	3	0.2	0.07	4.2	3	0.1	GW
At5	Intercept comms	0.25	7	5	0.35	0.23	3.15	5	0.14	SCADA
At6	Man in the middle	0.65	8	5	1.04	0.62	6.08	5	0.75	SCADA
At7	Spoof GW	0.15	4.5	4	0.17	0.07	2.25	4	0.04	GW
At8	Exploit Vuln in DB	0.45	7	3	1.05	0.23	2.8	3	0.21	DB
At9	Exploit Vuln in VM	0.9	7.5	4	1.69	0.36	4.5	4	0.41	VM_DB
At10	DB Account Hijacking	0.8	6.5	7.5	0.69	0.16	4.55	7.5	0.1	DB
At11	Read Data	0.6	8	4	1.2	0.12	7.2	4	0.22	DB

**Table 5 sensors-20-04404-t005:** Risk vectors for countermeasure nodes in the steal energy data ADT.

Defence Id	ADT Node Name	Defence Risk Vector				Asset
		Prob	Imp	Cost	Risk	
D1	RA-5_Vuln_scanning-1	0.6	6	7	0.51	GW
D2	AC-6_Least_privilege	0.4	4	3	0.53	GW
D3	SI-6_Security_and_Privacy_Function_Verification	0.5	5	5	0.5	GW
D4	SC-7_Boundary_Protection	0.3	7	6	0.35	GW
D5	SC-7.11_Boundary_Protection_Restrict_incoming_traffic	0.1	4.5	5	0.09	SCADA
D6	CA-8_Penetration_testing	0.25	4	5	0.2	SCADA
D7	SC-23_Session_Authenticity	0.15	6	4	0.23	
D8	SI-4_System_Monitoring	0.5	5	3	0.83	GW
D9	RA-5_Vuln_scanning-2	0.5	4	5	0.4	DB
D10	RA-5_Vuln_scanning-3	0.6	6	2	1.8	VM_DB
D11	AC-3_Access_Enforcement	0.8	7	8	0.7	DB
D12	SI-20_De-identification	0.8	7	7	0.8	DB

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
