# Peer review of "Continuous Quantitative Risk Management in Smart Grids Using Attack Defense Trees"

_sensors, 2020, doi:10.3390/s20164404_

Round 1

Reviewer 1 Report

This work mainly introduces a continuous quantitative risk management methodology for complex Smart Grid systems, which can help find the minimum optimal defence set that is able to cover all or a selected set of envisioned attack events.

Overall the idea is interesting and might be good for SG security. Some issues need to be justified.

- How to implement this method in practice, according to different requirements?

- A comparison with other risk analysis methods that can be used in SG, i.e., CORAS, NIST.

Reviewer 2 Report

1. Grammar and punctuation need corrections. The proposed changes are annotated in the attached pdf file.

2. At line 100, Attack Defence Trees (ADT) are mentioned without a bibliographic reference. I suggest the reference [9].

3. At line 115, the authors say:
"Attack Defence Trees (ADT) also known as Attack Countermeasure Trees (ACT) introduced by Kordy et al. [9] ..."

According to this sentence, ADT and ACT are the same type of model. In my knowledge, ADT and ACT have similarities, but are different types of models according to their definition in [9] and [16] respectively. So the above sentence should be changed, and reference [16] should be cited when mentioning ACT.

4. Is the ADT in fig. 2 drawn by means of ADTool?

If so, this should be said in the main text.

5. At lines 419-427, the operator & is used for AND, while the operator ∨ is used for OR. I suggest the use of the operator ∧ for AND, instead of &.

6. At line 517, the authors say:

"Mincuts or minimum cuts of ADT are different attack-defence suites that realise the main goal."

In my opinion, minimal cut sets (or mincuts) must be minimal sets of events causing the main goal. In other words, a minimal cut set must not contain other cut sets. So I suggest the replacement of "suites" with "minimal sets".

7. In table 5 and in the main text, the attack events are identified by at_i. I suggest the addition of such identifiers in the ADT in fig. 6.

8. Tab. 5 and its caption are in different pages. The caption should be close to the table, both in the same page.

9. The first two lines of section 4.2 have a font different from the rest of the paper.

10. At line 523 I read: "???? in T = 1".

It is not clear to me whether ????=1 or T=1. 

11. At line 536, does "OPT" mean "optimal" or is it an acronym?

12. The paper is quite long: 25 pages. If would be more readable if shorter.

Round 2

Reviewer 1 Report

No more comments.